# MULTI-INTEREST DISENTANGLED REPRESENTATION LEARNING FOR MULTIMODAL RECOMMENDATION

## ABSTRACT

In recent years, multimodal recommendation systems have been widely used in e-commerce and short video platforms. How to effectively utilize multimodal data and avoid the interference of multimodal noise information has become the key research direction of researchers. Many studies add multimodal data as auxiliary features to the model, which brings positive effects. Pictures, text and audio signals in short videos are more likely to attract users' interest than basic attributes. The user's multiple personalized interests largely determine the user's behavioral preferences. In order to effectively utilize user interest to improve model effect, We propose a new Multi-interest Disentangled Representation Learning method for multimodal recommendation (MIDR). Specifically, we first introduce the expected maximum to describe the relationship between interest and predicted target, and establish the optimization object based on multi-interest recommendation. Then, considering the relationship between user interest and multiple modalities, we introduce disentangled representation learning to extract modal sharing and modal specific interest representations. Furthermore, we introduce multi-interest contrast module to help model learning interest representation based on self-supervised learning. We conducted experiments on three real-world data sets, and our proposed MIDR outperformed other state-of-art models. The effectiveness of the disentangled interest representation module and interest contrast module was verified by the ablation experiment.

## 1 INTRODUCTION

In recent years, multi-modal recommendation system has been widely used, including short video platform, e-commerce platform and other applications covering rich multi-modal information (Deldjoo et al. (2020); Cen et al. (2020)). Early collaborative filtering algorithms mainly learn the implicit matching relationship by using the historical interaction between users and items through matrix decomposition and factorization (Koren et al. (2009); Rendle et al. (2012)). Then the depth model constructs the matching relationship between user and item based on the id embedding and attribute features. The emergence of multi-modal data provides the model with more abundant information, which can help the model to provide more accurate recommendations. Therefore, how to make full use of multimodal data effectively and avoid the interference of multimodal noise information has become the key research direction of researchers (Xu et al. (2020); Liu et al. (2019); Han et al. (2022)).

Some researchers use multimodal data as an auxiliary feature to extract visual and semantic features from images and text via encoders (Chen et al. (2019); Deldjoo et al. (2021)). The primary results are obtained by using the multimodal feature method directly. Later, some research work begin to mine the link between user preferences and multimodal data, and to extract the information of interest contained in images and text by means of attention modeling and other methods (Tao et al. (2020)). VBPR (He & McAuley (2016)) uses visual features extracted from product images to mine users' visual preferences for images. Recently, some work based on graph neural network modeling multimodal data has achieved good results. MMGCN (Wei et al. (2019)) constructs a user-item bipart graph on each mode and enriches the representation of each node with the topology and features of its adjacent nodes. In addition, there has been some work to improve model effectiveness from the perspective of modeling intentions and semantic topics by learning about implicit user preferences (Guo et al. (2022); Chen et al. (2022)).

However, there are some problems with these studies. First, it is difficult to learn an accurate semantic representation based on various modal data. Multimodal data not only provides new modal information, but also contains a lot of noise information. For example, user reviews contain ambiguous comments and images contain a large number of irrelevant elements. On the one hand, the model is easily disturbed by such noisy information, on the other hand, it is difficult for the model to judge which information is more important. When these noise-interfered and semantically ambiguous modal representations are added to the model as side information, it is easy for the model to learn biased estimation results and damage the long-term stability of the recommendation system.

Secondly, users should have personal interests and preferences when buying goods and watching short videos . For example, when buying clothes, users are not interested in the style, but prefer the color of clothes. Rather than being influenced by all the information about the product, users are more likely to act out of their own interest. Third, much of the work involved encoding multimodal data and then directly linking modal representations to attribute feature representations to learn higher-order interactions. The common assumption is that modal and attribute representations reside in the same semantic space. However, in practice, it is difficult to ensure that modal representation and attribute representation are aligned in semantic space.

Considering the above problems, we propose a new solution. We propose a new Multi-interest Disentangled Representation Learning method (MIDR) for Multimodal Recommendation. Specifically, we first introduce expected maximum to describe the relationship between interest and predicted target, and establish a multi-interest based recommendation framework. Then, considering the existence of shared interest and specific interest among multiple modes, we introduce disentangled representation learning to extract modal shared interest representation and modal specific interest representation. Then, we introduce the multi-interest contrastive module to construct the contrast relationship between interests, so as to help the model improve the learning ability of interest representation. Finally, the model is trained under the joint action of optimization objectives of multiple tasks.

In summary, the main contributions of this paper are as follows:

- We establish the relationship between multiple interests and recommended goals through expectation maximization, and introduce disentangled representation to learn modal shared interest representation and modal specific interest representation.

- We propose a multi-interest disentangled representation learning method (MIDR) for Multimodal Recommendation. It consists of multi-interest disentangled representation and multi-interest contrast learning module. MIDR not only accurately establishes an interest-based recommendation framework, but also realizes effective multimodal knowledge utilization.

- We conducted a comprehensive experiment on three publicly baseline data sets, and the experimental results show that our model has the best performance. Further ablation experiments and case studies have verified the validity of our proposed interest modeling. We will then release our code to the community.

## 2 RELATED WORK

In recent years, a lot of research work has been proposed to improve the effectiveness of multimodal recommendation (Du et al. (2022); Yu et al. (2022); Han et al. (2022)). Since the traditional collaborative filtering method cannot meet the requirements of large-scale multi-modal recommendation system, the method of modeling modal information using deep neural network has been developed rapidly. Some early work focused on multimodal data encoding, designing different encoders to extract visual and textual information, and achieved certain results (Chen et al. (2019); Deldjoo et al. (2021)). VBPR (He & McAuley (2016)) extracts visual features from product images and incorporates visual signals into a matrix decomposition model. VECF (Chen et al. (2019)) also models the user's attention perception information for different areas of the image and review. Due to the powerful representation ability of graph neural network, many works using graph neural network to utilize multimodal information have achieved good results (Zhao & Wang (2021); Yu et al. (2022)). MMGCN (Wei et al. (2019)) constructs a user-item dichotomous graph and enriches the representation of each node with the topology and characteristics of its adjacent nodes. GRCN (Wei et al.

(2020)) explores the impact of implicit feedback on GCN based recommendation models and proposes a structure that can adaptively improve user project interaction diagrams. MGAT (Tao et al. (2020)) transmits information in a single graph, and uses the gated attention mechanism to identify the different importance scores of different patterns on user preferences.

With the development of recommendation system, multi-interest recommendation has become one of the important research directions (Zheng et al. (2022b); Dhelim et al. (2020); Feng et al. (2019)). Many studies have proposed interest-based recommendation schemes and achieved certain results in the field of sequential recommendation. MIND (Li et al. (2019)) extracts multiple interests of users based on capsule network and dynamic routing algorithm combined with historical behaviors of users. ComiRec (Cen et al. (2020)) proposes a multi-interest module to capture multiple interests from a sequence of user behaviors, which can be used to retrieve candidate items from a large pool of items. ICL (Chen et al. (2022)) clusters the user's interest representation, and then represent the user's intention with the cluster center. The loss function is constructed according to the user's intention and the user's interest representation. CMI (Li et al. (2022)) builds multi-interest encoders based on implicit categories of items and proposes that contrast multi-interest loss minimizes the difference between interests extracted from two enhanced views of the same interaction sequence. MMDIN (Yang et al. (2021)) designs multi-head attention module to extract multi-modal information to improve the effectiveness of the recommendation system. CLSR (Zheng et al. (2022b)) uses contrastive learning to construct self-supervised learning task for long and short interests to show the differential modeling of long and short interests. By combining multi-interest learning and graph convolution aggregation, MGNM (Tian et al. (2022)) achieves a better modeling effect for users' multi-grained interest. HUIGN (Wei et al. (2021a)) presents user intentions in a hierarchical graph structure from fine to coarse-grained.

## 3 PRELIMINARIES

Let $U = \{u_1, u_2, \cdots, u_m\}$ and $I = \{i_1, i_2, \cdots, i_n\}$ be the sets of users and items respectively, where $m$ is the number of users, and $n$ is the number of items. $R \in \{0, 1\}^{m \times n}$ is the user-item implicit feedback matrix. Besides user-item interactions, multimodal features are offered as content information of items. We denote the modality features of item $i$ as $e_m^i \in R^{d_m}$, where $d_m$ denotes the dimension of the features, $m \in M$ is the modality, and $M$ is the set of modalities. The purpose of multimedia recommendation is to accurately predict users' preferences by ranking items for each user according to predicted preference scores $p$. In this paper, we consider visual, textual and acoustic modalities denoted by $M = \{v, t, a\}$.

## 4 THE PROPOSED MIDR MODEL

In this section, we mainly introduce the structure of MIDR, which consists of three parts. Firstly, we introduce the expected maximum to describe the relationship between the interest and the predicted goal, and establish the multi-interest recommendation optimization goal. Secondly, we introduce disentangled representation learning to extract modal shared interest representations and modal specific interest representations. Thirdly, we design an interest contrastive module to construct contrastive learning objectives to help the model learn accurate interest representation. Finally, the model is trained through the joint optimization of multiple tasks.

### 4.1 MULTI-INTEREST RECOMMENDATION THEORETICAL ANALYSIS

As we all know, user behavior is often the result of multiple interests. A user may be interested in fashion as well as art. When users visit the e-commerce platform, they will choose products according to their various interests. Therefore, modeling user interest may be a more effective way than directly using noise information and redundant information.

In the multimodal recommendation, the user's interest becomes richer because of the rich visual signals, textual signals and auditory signals received by the user. In particular, users have many different interests in the same modal data. For example, a female user might be interested in both the style and color of the clothes shown in the picture. Therefore, it makes sense to model the multiple interests of a user's multimodal perception.

It is assumed that there are $K$ interests $s^m = \{s_1, s_2, \cdots, s_K\}$ affecting the decision of the user in the modal m semantic space. In the multimodal scenario, each modal information can be of interest to the user. Therefore, the probability of a user interacting with a particular item can be expressed as:

$$P_w(z^u) = E_s[\sum_{m=1}^{M} P_w(z_m^u, s|m)] \tag{1}$$

Since we cannot directly observe interest, we can only define it by means of implicit representation. Assuming that every interest $s_i^m$ is an implicit representation, we can define the goal of model learning as:

$$w^* = \underset{w}{argmax}\, lnE_{(s)}[\sum_{m=1}^{M} P_w(z_m^u, s|m)] \tag{2}$$

Considering that the objective function is difficult to optimize, we solve the lower bound of the formula and maximize the lower bound for approximate solution. Suppose interest $s^m$ from distribution $D_s^m$, meet the condition $\sum_{s^m} D_{s_I^m} = 1$, and $D(s_I^m) \geq 0$. The following derivation can then be obtained:

$$\sum_{u=1}^{N} lnE_{(s)}[\sum_{m=1}^{M} P_w(z_m^u, s|m)] = \sum_{u=1}^{N} ln\sum_{i=1}^{K}[\sum_{m=1}^{M} P_w(z_m^u, s_i|m)] \tag{3}$$

Further, the above equation is equivalent to the following:

$$\sum_{u=1}^{N} ln\sum_{i=1}^{K}[\sum_{m=1}^{M} P_w(z_m^u, s_i|m)]$$
$$= \sum_{u=1}^{N} ln\sum_{i=1}^{K}[\sum_{m=1}^{M} Z(s_i|m)\frac{P_w(z^u, s_i|m)}{Z(s_i|m)}] \tag{4}$$

It is difficult to directly optimize the above formula, so we introduce music inequality to transform:

$$\sum_{u=1}^{N} ln\sum_{i=1}^{K}[\sum_{m=1}^{M} Z(s_i|m)\frac{P_w(z^u, s_i|m)}{Z(s_i|m)}]$$
$$\geq \sum_{u=1}^{N}\sum_{i=1}^{K}\sum_{m=1}^{M} Z(s_i|m)ln\frac{P_w(z^u, s_i|m)}{Z(s_i|m)} \tag{5}$$

Then, the formula proportional to the above formula can be expressed as:

$$\sum_{u=1}^{N}\sum_{i=1}^{K}\sum_{m=1}^{M} Z(s_i)lnP_w(z^u, s_i|m) \tag{6}$$

The above formula represents the lower bound of the model learning objective. However, since $D(s)$ cannot be directly observed, we still cannot directly optimize the above equation. Therefore, we need to learn approximate interest distribution through interest representation module.

## 4.2 MULTI-INTEREST DISENTANGLED REPRESENTATION

In the actual recommendation scenario, there may be sharing or orthogonal relationship between the user's interest in different modes. For example, if the user finds "British Brock carving" in the product description and the Brock carving pattern in the product picture, then the user's interest in

different modes is similar. On the contrary, if the user is interested in the material of the product in the description of the product and likes the female model in the picture, then the user's interest in different modes is irrelevant.

In order to describe modal sharing and model-specific interest representation more deeply, we assume that the distribution function $D_s^m$ is a mixed distribution composed of a modal sharing distribution and a modal specific distribution, which is defined as follows:

$$Z(c_i) = Z(c_i|r)P(r) + Z(c_i|n)P(n) \tag{7}$$

where $Z(c|r)$ represents a modal sharing distribution function and $Z(c|n)$ represents a modal specific distribution function. $P(r)$ represents the probability of modal correlation, and $P(n)$ represents the probability of modal independence, satisfying $P(r) + P(n) = 1$. To decouple modal sharing and modal - specific interest information, we use mutual information theory to model.

### 4.2.1 MUTUAL INFORMATION THEORY

We give the details of Mutual Information Theory in Appendix D. Let $X$ and $Z$ represent the two random variables, the mutual information maximization object between $X$ and $Z$ is defined as follows:

$$L_{\theta,\varphi}(X, Z) = \hat{I}^{JSD}(X, Z) \tag{8}$$

where Z is the representation obtained by an encoder with the parameter $\varphi$.

### 4.2.2 MODAL-SHARING INTEREST REPRESENTATION

Let $E_\varphi : X \to S_X$ represent a modal sharing interest representation encoder extracted from modal $X$, and $E_\varphi : Y \to S_Y$ represent a modal sharing interest representation encoder extracted from modal $Y$. For modal $X$, we hope to extract the sharing interest representation $S_X$ among the modalities, so that $S_X$ can reconstruct information close to $Y$. We estimate and maximize the mutual information between modal features and their sharing interest representations. The corresponding mutual information loss function is expressed as follows:

$$L_{m1} = L_{\theta,\varphi}(X, S_Y) + L_{\theta,\varphi}(Y, S_Z) \tag{9}$$

In addition, since we extract modal sharing interest representations from different modal data, these representations should theoretically be similar. Therefore, L2 distance is added to constrain the representation of modal sharing interest, defined as follows:

$$L_2 = E_{p(s_x, s_y)}[||S_X - S_Y||_2] \tag{10}$$

The objective function of extracting modal sharing representation consists of mutual information maximum and interest constraint, which is defined as follows:

$$L_{share} = \alpha L_{m1} - \gamma L_2 \tag{11}$$

### 4.2.3 MODAL-SPECIFIC INTEREST REPRESENTATION

Let $E_\omega : X \to G_X$ represent a modal specific interest representation encoder extracted from modal $X$, and $E_\omega : Y \to G_Y$ represent a modal specific interest representation encoder extracted from modal $Y$. To solve these representations, we estimate and maximize the mutual information between each modal feature and its corresponding interest representation $R$. $R$ is composed of modal sharing and modal specific interest representation, i.e., $R_X = (S_X, E_X)$. We hope that the original feature information can be reconstructed through modal sharing representation and modal specific representation. Mutual information loss function based on information reconstruction is defined as follows:

$$L_{m2} = L_{\theta,\varphi}(X, R_Y) + L_{\theta,\varphi}(Y, R_Z) \tag{12}$$

While maximizing mutual information between $X$ and $R_X$, $E_X$ should not contain information already captured by $S_X$, so it is necessary to minimize mutual information between $E_X$ and $S_X$, defined as follows:

$$minimize \quad I(E_X, S_X) \tag{13}$$

However, while maximizing the mutual information above, minimizing the mutual information of $E_X$ and $S_X$ is not convergent for the model. Therefore, an adversarial network is introduced to minimize the mutual information between $E_X$ and $S_X$ for the convenience of solving the model. Firstly, the modal data is passed through the encoder $E_{\varphi X}$ to extract the real samples satisfying the $P_{S_X E_X}$ distribution. Fake samples satisfying the $P_{S_X} P_{E_X}$ distribution are then extracted by shuffling the exclusive representation of samples within a batch. A discriminator $D_{\rho X}$ is then used to identify the real and fake samples extracted above. Therefore, we can achieve the minimization of mutual information between $E_X$ and $S_X$ by minimizing Jensen-Shannon divergence $D_{JS}(P_{S_X E_X} || P_{S_X} P_{E_X})$. The loss function based on adversarial network is defined as follows:

$$
\begin{aligned}
L_{adv} = &E_{p(s_x)p(e_x)}[log(D_{\rho X}(S_X, E_X))] \\
&+ E_{p(s_x, e_x)}[log(1 - D_{\rho X}(S_X, E_X))]
\end{aligned}
\tag{14}
$$

Further, by integrating the reconstruction loss function with the adversarial loss function, we can obtain the mode-specific learning objective as follows:

$$L_{specific} = \beta L_{m2} - \lambda \sum_m (L_{adv}^m) \tag{15}$$

where $m$ denotes different modalities.

### 4.2.4 Interest distribution representation

For each sample, we can obtain the shared interest representation $s_i^m$ and the specific interest representation $e_i^m$ in each mode. Therefore, the interest representation $h_i^m = [s_i^m, e_i^m]$ for each mode can be obtained. In order to obtain the weights of the interest distribution, we introduce an attention network to learn the relationship between different interest representations. From the perspective of efficiency and effectiveness, we introduce a SENET network with softmax as the encoder. $K$ interest representations $h_i^m$ are spliced together to learn a weight vector with output dimension $K$ through SENET. The representation of interest distribution is defined as follows:

$$\pi = softmax(SENET([s_1^m, s_2^m, \cdots, s_K^m])) \tag{16}$$

where $\pi \in R^K$ denotes the weight of interest distribution in the semantic space of modal m.

In addition, we believe that the conditional independence hypothesis is satisfied between different interests. In order to ensure the independence of different interests, we designs an orthogonal loss function for constraint, which is defined as follows:

$$L_c = \sum_{m=1}^{M} \sum_{i=1}^{K-1} \sum_{j=i+1}^{K} (s_i^T s_j)^2 \tag{17}$$

### 4.3 Multi-interest Contrast Learning

We assume that each intention distribution is an independent Gaussian distribution, then we can get:

$$P_w(z^u, s_i | m) = P_w(s_i | m) P_w(z^u | s_i, m) = \frac{1}{K} P_w(z^u | s_i, m) \tag{18}$$

The above formula is approximate as follows:

$$\frac{1}{K} P_w(z^u | s_i) \propto \frac{1}{K} \frac{exp(z^u s_i)}{\sum_{j=1}^{K} exp(z^u s_j)} \tag{19}$$

Thus, maximizing the likelihood function is equivalent to minimizing the contrast loss function as follows:

$$L_{cl} = -\sum_{u=1}^{N} \sum_{m=1}^{M} \sum_{i=1}^{K} \pi_i^u log \frac{exp(z^u s_i)}{\sum_{j=1}^{K} exp(z^u s_j)} \tag{20}$$

Table 1: Statistics of the three datasets with multimodal item Visual(V), Acoustic(A), Textual(T) information.

| Dataset | User | Item | Interactions | Embedding Dim | Sparsity |
|---------|------|------|--------------|---------------|----------|
| Tiktok | 9,319 | 6,710 | 59,541 | V(128), T(768), A(128) | 99.904% |
| Amazon-Sports | 35,598 | 18,357 | 256,308 | V(4,096), T(1024) | 99.961% |
| Amazon-Baby | 19,445 | 7,050 | 139,110 | V(4,096), T(1,024) | 99.899% |

### 4.4 TRAINING OPTIMIZATION

This paper focuses on the click prediction problem. The loss function is defined as follows:

$$L_p = -\frac{1}{N} \sum_i [y_i log p_i + (1 - y_i) log(1 - p_i)] \tag{21}$$

where p denotes the prediction result, which is obtained through a three-layer MLP based on interest representations $h^m$. $y$ represents the true label. The final loss function is defined as follows:

$$Loss = L_p + L_{share} + L_{specific} + \eta L_{cl} + \mu L_c \tag{22}$$

where $\lambda, \eta, \mu$ represent the hyperparameters used to control the effects of different losses. By optimizing the total loss function, the network parameters can be effectively updated.

## 5 EXPERIMENTS

In this section, we conduct extensive experiments to answer the following questions:

- **RQ1** How does our MIDR model perform compared to the state-of-the-art methods?
- **RQ2** How do different modules of MIDR contribute to the effectiveness of the model?
- **RQ3** How do the key parameters in the model affect the model effect?

### 5.1 EXPERIMENTAL SETTINGS

#### 5.1.1 DATASETS

We selected three widely used data sets for the experiment. The first dataset is Tiktok, extracted from the tiktok platform. Tiktok is a short video application with hundreds of millions of users, which contains a lot of rich multi-modal data and user information. The other two data sets were selected from the Amazon dataset[1], sports and baby. The statistical results of the three datasets after preprocessing are shown in Table 1. The details of datasets are provided in Appendix B.

#### 5.1.2 BASELINES

To evaluate the performance, we compared the proposed MIDR with the following baselines: **Light-GCN** (He et al. (2020)), **VBPR** (He & McAuley (2016)), **MMGCN** (Wei et al. (2019)), **GRCN** (Wei et al. (2020)), **SGL** (Kim et al. (2016)), **LATTICE** (Zhang et al. (2021)), **CLCRec** (Wei et al. (2021b)), **MMGCL** (Yi et al. (2022)), **HCGCN** (Mu et al. (2022)), **SLMRec** (Tao et al. (2022)), **MMSSL** (Wei et al. (2023)). The details of baselines are provided in Appendix A.

#### 5.1.3 EVALUATION METRICS AND PARAMETER SETTINGS

Following relevant work (Mu et al. (2022); Wei et al. (2023)), we adopt three commonly used evaluation metrics of recommendation systems for model evaluation in this paper, including Recall@K (R@K), Precision@K (P@K) and Normalized Discounted Gain (N@K). The learning rate is adjusted from [0.00001, 0.00005, 0.0001, 0.0005]. The dimension of the hidden vector is selected from [16, 32, 64, 128, 256]. The hyperparameters $\alpha$, $\gamma$, $\eta$ and $\mu$ are searched in [0.0001, 0.001, 0.01, 0.1, 1]. In addition, the number of interests is searched from [10, 20, 30, 50, 100].

---

[1]http://jmcauley.ucsd.edu/data/amazon/

Table 2: Overall performance comparison. Improvement denotes the relative improvements over the best baselines.

| Model | Amazon-Sports | | | Amazon-Baby | | | Tiktok | | |
|---|---|---|---|---|---|---|---|---|---|
| | R@20 | P@20 | N@20 | R@20 | P@20 | N@20 | R@20 | P@20 | N@20 |
| VBPR | 0.0582 | 0.0031 | 0.0265 | 0.0486 | 0.0026 | 0.0213 | 0.0380 | 0.0018 | 0.0134 |
| LightGCN | 0.0782 | 0.0042 | 0.0369 | 0.0698 | 0.0037 | 0.0319 | 0.0653 | 0.0033 | 0.0282 |
| MMGCN | 0.0638 | 0.0034 | 0.0279 | 0.064 | 0.0032 | 0.0284 | 0.0730 | 0.0036 | 0.0307 |
| GRCN | 0.0833 | 0.0044 | 0.0377 | 0.0754 | 0.0040 | 0.0336 | 0.0804 | 0.0036 | 0.0350 |
| LATTICE | 0.0915 | 0.0048 | 0.0424 | 0.0829 | 0.0044 | 0.0368 | 0.0843 | 0.0042 | 0.0367 |
| CLCRec | 0.0651 | 0.0035 | 0.0301 | 0.061 | 0.0032 | 0.0284 | 0.0621 | 0.0032 | 0.0264 |
| MMGCL | 0.0875 | 0.0046 | 0.0409 | 0.0758 | 0.0041 | 0.0331 | 0.0799 | 0.0037 | 0.0326 |
| SGL | 0.0779 | 0.0041 | 0.0361 | 0.0678 | 0.0036 | 0.0296 | 0.0603 | 0.0030 | 0.0238 |
| SLMRec | 0.0829 | 0.0043 | 0.0376 | 0.0765 | 0.0043 | 0.0325 | 0.0845 | 0.0042 | 0.0353 |
| MMSSL | 0.0998 | 0.0052 | 0.0470 | 0.0962 | 0.0051 | 0.0422 | 0.0921 | 0.0046 | 0.0392 |
| HCGCN | 0.1032 | 0.0055 | 0.0478 | 0.0922 | 0.0048 | 0.0415 | 0.0935 | 0.0049 | 0.0412 |
| MIDR | **0.1096** | **0.0059** | **0.0505** | **0.1033** | **0.0055** | **0.0457** | **0.0967** | **0.005** | **0.0425** |
| Improvement | 6.17% | 6.52% | 5.74% | 7.33% | 7.51% | 8.29% | 3.46% | 2.85% | 3.27% |

Table 3: Ablation study of MIDR.

| Model | Sports | | Baby | | Tiktok | |
|---|---|---|---|---|---|---|
| | R@20 | N@20 | R@20 | N@20 | R@20 | N@20 |
| MIDR | **0.1096** | **0.0505** | **0.1033** | **0.0457** | **0.0967** | **0.0425** |
| w/o-DR | 0.0852 | 0.0394 | 0.7627 | 0.0341 | 0.0823 | 0.0347 |
| w/o-IC | 0.0864 | 0.0391 | 0.7706 | 0.0335 | 0.0807 | 0.0352 |
| w/o-OC | 0.0993 | 0.0475 | 0.0944 | 0.0418 | 0.0908 | 0.0379 |
| r/p-IR | 0.0768 | 0.0354 | 0.0681 | 0.0322 | 0.0615 | 0.0336 |

## 5.2 OVERALL PERFORMANCE (RQ1)

We conducted comprehensive experiments on three data sets to compare the effects of our proposed MIDR with other advanced models. The experimental results are shown in Table 2. According to the experimental results, we can observe the following phenomena and conclusions. The proposed MIDR model outperforms all other models on the three data sets. The model effect does not change with data sparsity. MIDR establishes reliable associations between user interests and recommendation goals, and accurately extracts model-perceived shared and interests-specific representations through disentangled representation. Therefore, MIDR provides significant improvements across multiple data sets. Compared with self-supervised learning methods such as MMSSL, the proposed MIDR performance also outperformed them on all data sets. MIDR is better than classical contrast learning methods because it establishes an accurate representation of disentangled interest.

## 5.3 ABLATION EXPERIMENTS (RQ2)

Further, we conduct ablation experiments separately for each important module. ($a$) We remove disentangled representation (DR) from the model and directly use randomly generated vectors as interest representations. ($b$) We remove the multiple interest contrast (IC) loss function from the final optimization objective. ($c$) We remove the interest orthogonal constraint (OC) from the final optimization objective. ($d$) We remove the multi-interest recommendation (IR) framework established in this paper and directly use an encoder to learn multiple interest representations. We conducted a comprehensive ablation experiment on three data sets. The experimental results are shown in Table 3. According to the experimental results, we can observe the following phenomena and conclusions: When we replace the interest representation of disentangled representation with random vector, the model effect decreases obviously. The experimental results verify the effectiveness of learning users' interests with disentangled representation. Without the proposed multi-interest recommendation framework, the model based on simple encoder learning interest is significantly worse. This also illustrates the difficulty of building accurate user interests directly from the encoder.

## 5.4 HYPERPARAMETER ANALYSIS (RQ3)

Since the model contains some hyperparameters which will affect the effect of the model, we carry out sensitivity analysis on some important parameters. **Impact of parameters $\alpha$ and $\gamma$.** Since we introduce the modal-sharing interest loss function, we explore the influence of loss weight $\alpha$ and $\gamma$ on the model effect. The two parameters are searched in [0.0001, 0.001, 0.01, 0.1, 1]. As shown in Fig. 1, with the gradual increase of the weight, the effect of the model presents a phenomenon of first improvement and then decline. Because these two parameters control the size of mutual information loss, the larger the value, the stronger the constraint. However, too much weight can easily lead to instability of model training and decrease the effect. In addition, we also adjusted parameters $\beta$ and $\lambda$, the phenomenon is similar to modal-sharing interest loss function. **Impact of embedding dimension $d$** As shown in Fig. 2, the effectiveness of the model is gradually improved with the increase of the embedding dimension. The range of effect improvement decreases with the further improvement of dimension. In the experiment, complexity should be fully considered to choose the appropriate dimension. **Impact of parameters $\eta$ and $\mu$** is provided in Appendix C, with the increasing of the weight, the effect of the model decreases gradually.

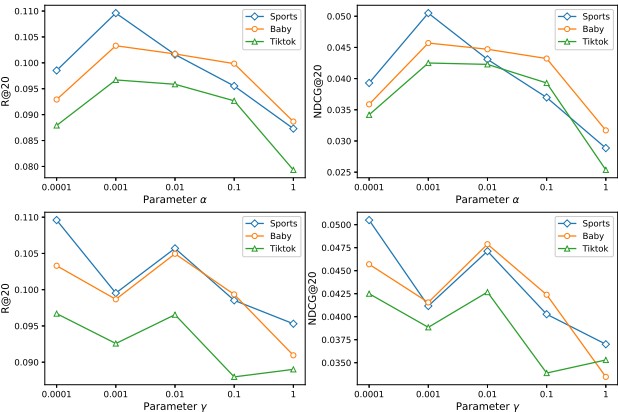

Figure 1: Impact of parameters $\alpha$ and $\gamma$.

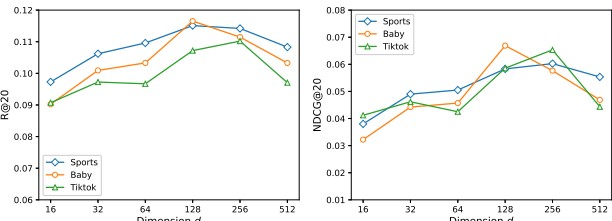

Figure 2: Impact of embedding dimension $d$.

## 6 CONCLUSION

In this paper, we propose a multi-interest disentangled representation learning method. MIDR effectively learns multimodal interest representation to provide accurate recommendations. On three widely used real data sets, the proposed MIDR outperforms other advanced models. The comprehensive experiment verifies the effectiveness of each module designed by us. In the future, we want to try to introduce interest modeling in a cross-domain multi-modal recommendation scenario. And we will explore more rich ways to utilize multimodal data such as the use of hypergraph.

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

# A    DETAILS OF THE BASELINES

To evaluate the performance, we compared the proposed MIDR with the following baselines:

- **LightGCN** (He et al. (2020)) simplifies the model while ensuring its effectiveness by preserving the operation of the most core aggregated neighbor nodes.

- **VBPR** (He & McAuley (2016)) utilizes attention mechanism to extract users' visual preferences for product images, and incorporates visual information into the matrix decomposition model.

- **MMGCN** (Wei et al. (2019)) believes that mining user preferences solely through user interaction history is not sufficient, and proposes a graph convolutional network to utilize multimodal data.

- **GRCN** (Wei et al. (2020)) explores the impact of implicit feedback on GCN based recommendation models and proposes a structure that can adaptively improve user project interaction diagrams.

- **SGL** (Kim et al. (2016)) has designed three different data enhancement operators to build comparative learning objectives based on graph collaborative filtering.

- **NCL** (Kim et al. (2016)) utilizes structural adjacent node information to generate positive sample pairs based on em clustering for constructing contrastive learning.

- **LATTICE** (Zhang et al. (2021)) learns the item to item structure for each modality and aggregates multiple patterns to obtain potential item graphs.

- **CLCRec** (Wei et al. (2021b)) constructs two effective contrastive learning optimization objectives based on user and interaction items, effectively alleviating the cold start problem.

- **MMGCL** (Yi et al. (2022)) uses modal edge loss and modal masking to generate user project diagrams, and introduces a new negative sampling technique to learn the correlation between modalities.

- **HCGCN** (Mu et al. (2022)) designs corresponding clustering losses to enhance user-item preference feedback and multimodal representation learning constraints to adjust modal importance.

- **SLMRec** (Tao et al. (2022)) designs three data augmentation methods for feature dimensions and proposed corresponding comparative learning objectives to optimize model training.

- **MMSSL** (Wei et al. (2023)) introduces cross-modal contrast learning method to maintain semantic commonality between modalities and diversity of user preferences.

# B    IMPLEMENTATION DETAILS

Following relevant work (Mu et al. (2022); Wei et al. (2023)), we adopt three commonly used evaluation metrics of recommendation systems for model evaluation in this paper, including Recall@K (R@K), Precision@K (P@K) and Normalized Discounted Gain (N@K). The learning rate is adjusted from [0.00001, 0.00005, 0.0001, 0.0005]. The dimension of the hidden vector is selected from [16, 32, 64, 128, 256]. The hyperparameters $\alpha$, $\gamma$, $\eta$ and $\mu$ are searched in [0.0001, 0.001, 0.01, 0.1, 1]. In addition, the number of interests is searched from [10, 20, 30, 50, 100]. Following the data processing method adopted by (Zheng et al. (2022a)), we select users who interact in both domains, and then filter users and items that interact less than 10 times. The visual features are provided by the data set and represented as 4096-dimensional embedding. Following (Mu et al. (2022)), we connect the item title, description and brand together to extract text features.

# C    SENSITIVITY ANALYSIS

**Impact of parameters $\eta$ and $\mu$.** Since the weights $\eta$ and $\mu$ control the effects of interest contrast loss, we adjust these two parameters to study the effects on the model. $\eta$ and $\mu$ are adjusted in [0.0001, 0.001, 0.01, 0.1, 1]. As shown in Fig. 3, with the increasing of the weight, the effect of

the model decreases gradually. Since these two parameters directly affect the learning of interest representation, it is easier for the model to train stably with a smaller weight, thus achieving better results.

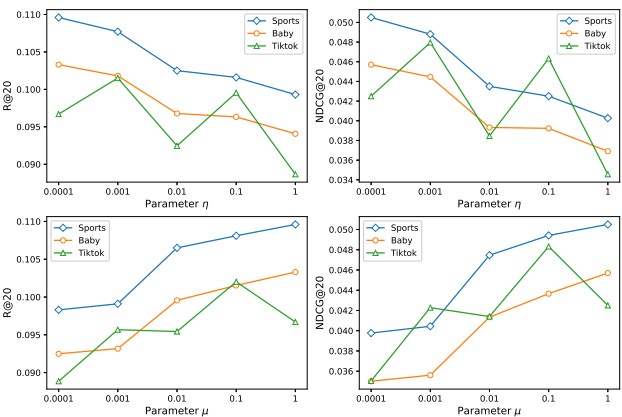

Figure 3: Impact of parameters $\eta$ and $\mu$.

## D  MUTUAL INFORMATION THEORY

Let X and Z represent the two random variables, p(x) and p(z) represent the corresponding marginal probability density function, and p(x, z) represent the joint probability density function of the two. Then the mutual information expression of X and Z is as follows:

$$I(X, Z) = \int_Z \int_Z p(x, z) log(\frac{p(x, z)}{p(x)p(z)}) dx dz \tag{23}$$

The above equation can theoretically be rewritten as $I(X, Z) = D_{KL}(P(_{XZ}||P_X P_Z)$, where $P_{XZ}$ represents the joint probability distribution of $X$ and $Z$, and $P_X$ and $P_Z$ represent the edge distributions. Following Deep InfoMax, we adopt Jensen-Shannon divergence as the objective function, which proved to be very stable. This leads to the following object:

$$\hat{I}^{JSD}(X, Z) = E_{p(x,z)}[-log(1 + e^{-T_\theta(x,z)})]$$
$$- E_{p(x)p(z)}[log(1 + e^{T_\theta(x,z)})] \tag{24}$$

where $T_\theta$ denotes a representation neural network. Further, the mutual information maximization object between $X$ and $Z$ is defined as follows:

$$L_{\theta,\varphi}(X, Z) = \hat{I}^{JSD}(X, Z) \tag{25}$$

where Z is the representation obtained by an encoder with the parameter $\varphi$.

