# OpenReview forum: "Multi-interest Disentangled Representation Learning for Multimodal Recommendation"
_ICLR.cc/2024/Conference — Submitted to ICLR 2024_

### Official Review · Reviewer_2Vun · 2023-10-30

**Soundness:** 2 fair
**Presentation:** 3 good
**Contribution:** 2 fair
**Rating:** 5
**Confidence:** 4

**Summary:**

In this work, the authors focus on the multimodal recommendation system. Considering the challenge of effectively using multimodal data the authors proposed to mitigate its noise by implementing a Multi-interest Disentangled Representation Learning (MIDR). In particular, they optimize multi-interest recommendations through disentangled interest representation and self-supervised learning. Experiment results on real-world datasets demonstrate MIDR's superior performance compared to existing models.

**Strengths:**

1. The authors analysed the drawback of multi-modal information in the recommender system.
2. They authors proposed a new method to disentangle the multi-interests in the item representation learning.
3. By conducting the experiments on public datasets, the authors demonstrated the effectiveness of their model.

**Weaknesses:**

1. The novelty of the proposed model is limited. From my reading, the authors just combined the existing modules together, such as contrastive learning and deep learning.
2. I am not sure how the authors implement their proposed model. From the manuscript, the authors seem to derive the objective of their model, but ignoring the detail of implementation. It is hard to reproduce the proposed model.

**Questions:**

Please refer to the weaknesses.

---

> ### Author Response · Authors · 2023-11-23
> **Reply to Reviewer 2Vun**
>
> Thank you very much for your careful review and your valuable comments.
>
> In response to your question, we make the following reply:
>
> 1. As you can see, our analysis of ablation experiments shows that the various components of the model are valid. This just happens to illustrate the overall effectiveness of our approach. Although the individual modules we use may seem to already exist, our overall design idea is novel. First of all, the multi-interest-based theoretical analysis and derivation in Section 4.1 directly established the connection between the user preference factor we considered (interest) and the goal we want to optimize (accurate matching). Based on such a strong theoretical connection, we carried out the design of the following parts. In Section 4.2, we present a multi-interest decoupling representation method based on conditional probability, which leads to the effective learning of modal-sharing and modal-specific interest representation. Then in section 4.3, it appears to be a contrast learning, but it is actually a goal related to the optimized lower bound estimation we defined in Section 4.1, rather than a simple contrast learning. It is precisely through this analysis and derivation that we conclude that the lower bound of loss based on interest prediction can be approximated by the method of comparing the form of learning objectives. Therefore, these are our core ideas and innovation points. I hope you can reconsider our design ideas.
>
> 2. We can publish our implementation details at any time if needed. Due to the limited manuscript pages and the fact that we were rushing the paper, we did not add code to the appendix. But if we're allowed, we can upload our code.

---

### Official Review · Reviewer_MGCH · 2023-10-31

**Soundness:** 2 fair
**Presentation:** 2 fair
**Contribution:** 2 fair
**Rating:** 3
**Confidence:** 4

**Summary:**

The paper proposes a Multi-Interest Disentangled Representation Learning method for multimodal recommendation systems. The technique involves disentangled representation learning and a multi-interest contrast module to extract modal sharing and modal-specific interest representations. The proposed method outperforms other advanced models on three widely used real datasets. The paper's main contributions are the proposed method, establishing the relationship between multiple interests and recommended goals and the comprehensive experiment on three publicly baseline datasets.

**Strengths:**

1. The problem of concern in this paper has a practical background and is reasonably motivated. It has research value.
2. The structure of the paper is clear, the logic is smooth, and the proposed method is clearly demonstrated.
3. Extensive experiments are carried out on three datasets with rich baseline models.

**Weaknesses:**

1. The author's motivation is reasonable, but it is not well summarized. The motivation in the text is somewhat vague and not specific enough. On top of that, how the different components in the model correspond to the motivation is not well specified. The method is not closely related to the motivation.
2. The method in Section 4.2.3 is confusing. It is not reasonable for modal-specific interest representation and R to maximize mutual information, because S_X in R is modal shared. The authors later additionally minimize mutual information modal specific interest representation and S_X. This raises the question of why to maximize in the first place.
3. There are a lot of notation errors, and I guess in section 4.2.3 the author wanted to use E_X to mean the same thing as G_X. There is also a SZ in Section 4.2.2 that appears without explanation. These errors greatly affect the normal understanding of the method proposed by the authors.
4. The present method introduction and experimental results do not show that the method proposed by the author effectively solves the problems proposed in the motivation. The author should use more convincing explanations or design experiments to show this.
5. I suggest using images to visualize the model.

**Questions:**

1. How does the proposed approach correspond to the motivation? Is there a strong argument or proof that the method improves the correspondence problem, rather than simply improving the overall performance?
2. The method in Section 4.2.3 is confusing. It is not reasonable for modal-specific interest representation and R to maximize mutual information, because S_X in R is modal shared. The authors later additionally minimize mutual information modal specific interest representation and S_X. This raises the question of why to maximize in the first place.
3. The experimental data of w/o-OC and r/p-IR on the baby dataset in the ablation experiment are strange, which I feel may need an appropriate explanation.

---

> ### Author Response · Authors · 2023-11-23
> **Reply to Reviewer MGCH**
>
> Thank you very much for your careful review and your valuable comments.
>
> In response to your question, we make the following reply:
>
> 1. First of all, the multi-interest-based theoretical analysis and derivation in Section 4.1 directly established the connection between the user preference factor we considered (interest) and the goal we want to optimize (accurate matching). Based on such a strong theoretical connection, we carried out the design of the following parts. In Section 4.2, we present a multi-interest decoupling representation method based on conditional probability, which leads to the effective learning of modal-sharing and modal-specific interest representation. Then in section 4.3, it appears to be a contrast learning, but it is actually a goal related to the optimized lower bound estimation we defined in Section 4.1, rather than a simple contrast learning. It is precisely through this analysis and derivation that we conclude that the lower bound of loss based on interest prediction can be approximated by the method of comparing the form of learning objectives. Therefore, these are our core ideas and innovation points. As you can see, our analysis of ablation experiments shows that the various components of the model are valid.
>
> 2. We hope that the original feature information can be reconstructed through modal sharing representation and modal specific representation. As for the definition of mutual information loss, we adopt a general approach, similar to the DienCDR model, and our aim is to achieve information reconstruction. Because the greater the mutual information, the higher the correlation between the two variables, that is, the same key information is retained.
>
> 3. Since the baby dataset is the sparsest dataset, the training method for it needs to be more effective. Therefore, when we remove OC and IR, which can significantly affect the representation effect, the model effect has a relatively large decline.

---

### Official Review · Reviewer_NfjL · 2023-11-06

**Soundness:** 2 fair
**Presentation:** 1 poor
**Contribution:** 2 fair
**Rating:** 3
**Confidence:** 4

**Summary:**

The paper proposes a Multi-interest Disentangled Representation Learning method for multimodal recommendation (MIDR). The method aims to address different issues in multimodal recommendation, namely: 1) the existence of noisy information in multimodal content adopted for recommendation; 2) users’ decision may be driven by separate interest factors underlying each modality; 3) the assumption that each multimodal feature should be represented in the same semantic latent space may be wrong. After introducing and formalizing the multi-interest disentangled representation, the authors propose four additional components to the usual recommendation loss function (i.e., binary cross-entropy for the task of click prediction). The loss components account for: 1) modal-sharing interest representation, 2) modal-specific interest representation, 3) interest distribution representation and 4) multi-interest contrast learning. Extensive experimental settings when testing the model’s performance against eleven state-of-the-art (multimodal) recommender systems on three recommendation datasets show the efficacy of the MIDR solution. Moreover, an ablation study, complemented by a hyper-parameter values analysis, further motivate the architectural choices for MIDR.

**Strengths:**

+ The problem of multi-interest disentangled representation learning in multimodal recommendation is a hot topic in the field.
+ The experimental analysis is extensive with several evaluation dimensions.

**Weaknesses:**

---
### Before the rebuttal

- There are numerous conceptual flows in the formal presentation of the methodology. For instance, equation 1 introduces several terms and notions which have not been properly detailed in the preliminaries.
- From a general overview of the paper, it becomes hard to find the real novelty of the proposed approach, as it encompasses several winning approaches from the recent literature.
- The performance improvement with respect to the tested baselines seem a bit incremental. The authors may try to run statistical significance tests of MIDR performance with respect to the baselines.
- No code of the proposed MIDR approach is released at review time, which makes it difficult to further assess its efficacy and reproducibility.

---
### After the rebuttal

Dear Authors,

thank you for your rebuttal, which I carefully read.

Indeed, the ablation study shows the efficacy of each introduced module in terms of **recommendation performance**. However, even after considering your rebuttal, I still cannot see the **technical** novelty provided by the overall MIDR approach, which still appears to me as a combination of various winning approaches from the recent literature.

In this respect, I believe the methodology section from the paper is not helpful to clarify these aspects. As already outlined in my review, it is very hard to follow the formalism and methodology due to the numerous missing conceptual details in section 4, which are sometimes taken for granted.

Moreover, I am still convinced the performance improvement with respect to the other baselines is not quite evident, and it might require further statistical tests. Indeed, the sharing of code at review time would have been useful to test the effectiveness of the approach from an implementation point of view.

In light of above, I would keep the initial rating I gave to the paper. In my opinion, If all such aspects were addressed, the paper could be ready for submission to other venues. I strongly encourage the authors move towards this direction.

**Questions:**

---
### Before the rebuttal

* With reference to weakness 2, all model’s components are contributing to the overall performance; however, what is the main solution MIDR is proposing to the outlined issues?

---
### After the rebuttal

Please see the weaknesses section.

---

> ### Author Response · Authors · 2023-11-23
> **Reply to Reviewer NfjL**
>
> Thank you very much for your careful review and your valuable comments.
>
> In response to your question, we make the following reply:
>
> As you can see, our analysis of ablation experiments shows that the various components of the model are valid. This just happens to illustrate the overall effectiveness of our approach. Although the individual modules we use may seem to already exist, our overall design idea is novel. First of all, the multi-interest-based theoretical analysis and derivation in Section 4.1 directly established the connection between the user preference factor we considered (interest) and the goal we want to optimize (accurate matching). Based on such a strong theoretical connection, we carried out the design of the following parts. In Section 4.2, we present a multi-interest decoupling representation method based on conditional probability, which leads to the effective learning of modal-sharing and modal-specific interest representation. Then in section 4.3, it appears to be a contrast learning, but it is actually a goal related to the optimized lower bound estimation we defined in Section 4.1, rather than a simple contrast learning. It is precisely through this analysis and derivation that we conclude that the lower bound of loss based on interest prediction can be approximated by the method of comparing the form of learning objectives. Therefore, these are our core ideas and innovation points. I hope you can reconsider our design ideas.

---

### Official Review · Reviewer_UexV · 2023-11-11

**Soundness:** 3 good
**Presentation:** 2 fair
**Contribution:** 3 good
**Rating:** 6
**Confidence:** 4

**Summary:**

This paper proposes a new Multi-interest Disentangled Representation Learning method for multimodal recommendation (MIDR). Specifically, the author tries to combine multi-interest modeling and multi-modal modeling through mutual information theory for recommendation. The author tries to model the modal interest representation from two views: modal-sharing interest representation modeling and modal-specific interest representation modeling. Furthermore, the author introduces an orthogonal loss function for constraint and a multi-interest contrast module to help model learning interest representation based on self-supervised learning. The author further verifies the validity of the model by testing on three data sets.

**Strengths:**

This paper innovatively tries to combine multi-interest learning and multimodal learning to model a recommender system. The authors attempt to introduce two kinds of interest-based modeling, modal-shared and modal-specific, for each mode to enhance the model's extraction of valid information. The author constructs the mathematical representation of related concepts through rigorous mathematical reasoning, and gives the modeling steps in detail, and verifies the model effect on three data sets.

**Weaknesses:**

This paper is a little bit deficient in writing. For example, the article lacks an overview diagram that directly represents the proposed model, making it difficult for the readers to quickly understand the overall structure and characteristics of the model. Also, some key introductions are missing. For example, the symbols “z” and “w” in equation (1) and (2) are not clearly introduced. The “music inequality” in 4.1 (equation (5)) also lacks a brief introduction or citation. Moreover, the paper seems to use different symbols in some places to indicate the same or similar definition (e.g., “G” and “E” in 4.2.3; “R” and “h” in 4.2.3 and 4.2.4(I think it might be better to use “r” instead of “h”)), which puzzled me. In addition, there is too little analysis, comparison, and discussion of the overall experimental effect, which may make readers doubt the effect of the model.

**Questions:**

1.	In Section 5.4, why does the model effect decrease when the embedding dimension is 512? Logically, the desired situation should have a smaller improvement than 256.
2.	In Appendix C, could the hyperparameters of μ and η be experimented with smaller values to verify that they are actually beneficial to the model effect within a certain range? While the current results can be explained by stability, they can also be interpreted as hindering model performance.

---

> ### Author Response · Authors · 2023-11-23
> **Reply to Reviewer UexV**
>
> Thank you very much for your careful review and your valuable comments.
>
> In response to your question, we make the following reply:
>
> 1. In Section 5.4, when the embedding dimension is further increased to 512, the effect of the model will slightly decline, which is reasonable both in theory and in practice. Although the larger embedding dimension will guarantee a stronger representation capability in theory, the premise is that the model is fully trained and learned, which is difficult to guarantee. If we follow the logic that the model effect will be stronger if we increase the dimension, we only need to increase infinitely and the model effect will always improve, but this is not possible in practice. Limited by the quality of the data and the difficulty of the task, the model does not always become stronger with the increase of the representational dimension. In addition, in some of the comparison baselines mentioned in this paper, they also show similar experimental phenomena for dimension analysis.
>
> 2. In Appendix C, we can indeed try to set smaller values for the hyperparameters \mu and \eta to verify their impact on the model effect.  However, given that baselines' parameter analyses are typically performed in accordance with the parameters set in our paper, we were concerned that smaller or larger parameter Settings would be questioned by others as the result of our model's deliberate selection through multiple attempts, rather than its own capabilities. Therefore, we mainly refer to the similar parameter Settings of baselines. But we think your proposal is very good. We will supplement the effect of the model with lower hyperparameter values in the next version of the paper.

---

### Meta-Review · Area_Chair_fsgy · 2023-12-12

**Metareview:**

This paper received (slightly) contrasting ratings, 6, 3, 3, and 5, so mostly leaning to negative.
Among the positive comments, a reviewer assessed the paper to have a certain degree of innovation and a proper experimental analysis, but also recognized some difficulties in understanding given the poor presentation of the work, which is an issue also detected by other reviewers. Other remarks regard the incremental nature of the work, the unconvincing performance figures, needing a deeper statistical analysis to show the actual improvement wrt the state-of-the-art techniques, and lack of details to allow the reproducibility of the methodology. Overall, the limited novelty and the presence of many flaws and errors in the presentation, impacting the comprehension of the work, are the major problems detected for this paper.  The authors' rebuttal do not lead reviewers to change their scoring and evaluations, remaining mostly negative.

For these reasons, this paper is not considered acceptable for publication to ICLR 2024.

**Justification For Why Not Higher Score:**

Paper with lots of issues, not solved by the rebuttal. Even if there is a slightly positive review, it is not so positive to deserve a discussion for raising the evaluation.

**Justification For Why Not Lower Score:**

N/A

---

### Decision · Program_Chairs · 2024-01-16

Reject